# The Preparation and Catalytic Properties of Nanoporous Pt/CeO_2_ Composites with Nanorod Framework Structures

**DOI:** 10.3390/nano9050683

**Published:** 2019-05-02

**Authors:** Haiyang Wang, Dong Duan, Chen Ma, Wenyu Shi, Miaomiao Liang, Liqun Wang, Xiaoping Song, Lumei Gao, Zhanbo Sun

**Affiliations:** School of Science, MOE Key Laboratory for Non-equilibrium Synthesis and Modulation of Condensed Matter, Key Laboratory of Shaanxi for Advanced Functional Materials and Mesoscopic Physics, State Key Laboratory for Mechanical Behaviour of Materials, Xi’an Jiaotong University, Xi’an 710049, China; why0224@stu.xjtu.edu.cn (H.W.); duandong411@stu.xjtu.edu.cn (D.D.); machen@stu.xjtu.edu.cn (C.M.); swy890725@stu.xjtu.edu.cn (W.S.); lmm224@stu.xjtu.edu.cn (M.L.); wanglq@mail.xjtu.edu.cn (L.W.); xpsong@mail.xjtu.edu.cn (X.S.); lmgao@mail.xjtu.edu.cn (L.G.)

**Keywords:** dealloying, Pt/CeO_2_, nanorod framework, CO oxidation

## Abstract

Pt/CeO_2_ catalysts with nanoporous structures were prepared by the facile dealloying of melt-spun Al_92−X_Ce_8_Pt_X_ (X = 0.1; 0.3 and 0.5) ribbons followed by calcination. The phase compositions and structural parameters of the catalysts were characterized by X-ray diffraction (XRD), field emission scanning electron microscopy (FESEM) and high-resolution transmission electron microscopy (HRTEM). The specific surface area and pore size distribution were characterized by N_2_ adsorption–desorption tests. The catalytic properties were evaluated by a three-way catalyst (TWC) measurement system. The results revealed that the dealloyed samples exhibited a nanorod framework structure. The Pt nanoparticles that formed in situ were supported and highly dispersed on the CeO_2_ nanorod surface and had sizes in the range of 2–5 nm. For the catalyst prepared from the melt-spun Al_91.7_Ce_8_Pt_0.3_ ribbons, the 50% CO conversion temperature (T_50_) was 91 °C, and total CO could be converted when the temperature was increased to 113 °C. An X-ray photoelectron spectroscopy (XPS) test showed that the Pt_0.3_/CeO_2_ sample had a slightly richer oxygen vacancy; and a H_2_ temperature-programmed reduction (H_2_-TPR) test demonstrated its superior adsorption ability for reduction gas and high content of active oxygen species. The experiments indicated that the catalytic performance could be retained without any attenuation after 130 h when water and CO_2_ were present in the reaction gas. The favorable catalytic activities were attributed to the high specific areas and small pore and Pt particle sizes as well as the strong interactions between the CeO_2_ and Pt nanoparticles. The Pt nanoparticles were embedded in the surface of the CeO_2_ nanorods, inhibiting growth. Therefore, the catalytic stability and water resistance were excellent.

## 1. Introduction

The efficient catalytic oxidation of carbon monoxide is a very significant topic because of its widespread applications in CO gas sensors, indoor air purification, and industrial fields, including pollution control devices and automotive exhaust treatment [1,2]. Typically, noble metals and transition metals, such as Au [3,4], Ag [5], Pt [6], Pd [7], Rh [8], Co [9,10] and Cu [11], have been widely investigated as catalysts for CO oxidation. Among these metals, platinum stands out because of its high thermal stability [12]. However, the high cost of Pt greatly restricts its wide application. Furthermore, Pt-based catalysts are susceptible to sintering effects, which can lead to a decrease in the active area of the catalysts because of aggregation and coarsening [13,14,15]. Therefore, it is desirable to reduce the content of Pt in a catalyst and introduce metal oxides to construct Pt- and metal oxide-based heterostructures. Noble metals dispersed on specific metal oxides with high storage/release oxygen capacities are highly active towards catalytic CO oxidation.

Cerium dioxide (CeO_2_), as a representative and also one of the most abundant rate earth oxides on the planet, has unique heterogeneous catalytic abilities due to its excellent storage/release oxygen capacity and numerous oxygen vacancy defects [16]. CeO_2_ also provides an oxygen source through the fast and reversible Ce^4+^/Ce^3+^ redox reaction [17]. Therefore, based on these advantages, CeO_2_ has a wide range of potential applications and favorable development prospects in the field of air pollution. The morphology and facets of CeO_2_-based nanocomposites can greatly influence the formation and migration of surface oxygen vacancies. Also, nanosized structured CeO_2_ materials, including nanospheres, nanorods [18], nanocubes and nanotubes, have been synthesized [12,19,20,21]. Among these structures, nanorod-shaped CeO_2_ has received a substantial amount of attention because of its potentially large surface area and abundance of oxygen vacancy defects.

Because of its enhanced catalytic performance and wide range of applications, such as CO oxidation and methane partial oxidation, Pt/CeO_2_ has attracted increasing attention [22,23]. As the synthetic method has a large influence on the distribution of the oxygen vacancies and the structure as well as the ability to control the activity of a catalyst, a number of preparation methods, including the template, conventional impregnation, deposition–precipitation, and photodeposition methods, have been developed to synthesize Pt/CeO_2_ catalysts [24,25]. For example, Wei et al. prepared three-dimensional Pt@CeO_2_ nanoparticles through a colloidal crystal template method, which exhibited a high catalytic activity [26]. However, the impregnation and deposition–precipitation methods cannot effectively control the size and dispersion of the metals; additionally, the high-cost, sophisticated technology and low yield of the template method restricts its further development [27,28]. Dealloying, a facile method for fabricating nanoporous materials, has been attracting increased attention because of its green and effective preparation characteristics. Zhang et al. reported highly active Cu-doped CeO_2_ nanowires fabricated from a dealloying–annealing method for CO oxidation [29]. Our team has also performed research on CeO_2_, NiO/CeO_2_ and Ag/CeO_2_ nanorod frameworks synthesized by dealloying methods, which showed superior catalytic activity for CO oxidation due to their high porosities and large specific surface areas [30,31,32]. However, only a few works have reported the fabrication of Pt/CeO_2_ nanorods through a facile dealloying method for catalytic CO oxidation.

In this paper, a Pt/CeO_2_ nanorod framework catalyst for CO oxidation was fabricated by dealloying a melt-spun Al–Ce–Pt alloy followed by calcination. SEM and TEM characterization demonstrated that the Pt nanoparticles that formed in situ were supported and highly dispersed on the CeO_2_ nanorod surface, and an enhanced catalytic performance was obtained. For the Pt/CeO_2_ nanorods prepared from the Al_91.7_Ce_8_Pt_0.3_ precursor alloy, the 50% CO conversion temperature (T_50_) was 91 °C, and all of the CO could be converted when the temperature was increased to 113 °C. An excellent square curve for the degree of CO conversion was exhibited. The formation mechanism, microstructure and catalytic mechanism of the nanoporous Pt/CeO_2_ were studied.

## 2. Experimental

### 2.1. Preparation of the Pt/CeO_2_ Catalysts

As-cast Al_91.9_Ce_8_Pt_0.1_, Al_91.7_Ce_8_Pt_0.3_ and Al_91.5_Ce_8_Pt_0.5_ alloy button ingots were prepared from pure Al (99.90 wt%), pure Ce (99.90 wt%) and pure Pt (99.99 wt%) by arc-melting six times under a protective atmosphere of high-purity Ar. The ribbons were obtained after the alloy ingots were re-melted and rapidly solidified into foils through a melt-spinning technique under a high-purity Ar atmosphere; the resultant ribbons were 3–4 mm in width and 20–40 μm thick. Subsequently, the precursor alloy ribbons were dealloyed in a 20 wt% NaOH aqueous solution at room temperature for 2 h and then at 80 °C for 10 h. The resulting Pt/Ce–OH precursor nanorod framework-structured samples were carefully rinsed with deionized water several times. Finally, the dealloyed ribbons were calcined in a muffle furnace at 200 °C, 300 °C, 400 °C and 500 °C, respectively, for 2 h.

### 2.2. Material Characterization

The phase compositions of the dealloyed Al–Ce–Pt ribbons were collected by X-ray diffractometry (XRD-6100, Shinadzu Inc., Kyoto, Japan) with radiation from a Cu target (Kα, λ = 0.3615 nm). The morphologies and microstructures of the dealloyed ribbons were measured by field emission scanning electron microscopy (FESEM, JSM-7000F, JEOL Inc., Tokyo, Japan) and high-resolution transmission electron microscopy (HRTEM, JEM-2100, JEOL Inc., Tokyo, Japan). Energy dispersive spectroscopy (EDS) analysis and mapping were performed using scanning transmission electron microscopy (STEM, JEM-F200, JEOL Inc., Tokyo, Japan) equipped with an Oxford Instruments EDS spectrometer. An X-ray photoelectron spectroscopy (XPS, ESCALAB Xi+, Fisher Inc., Marshalltown, IA, USA) with Al-Ka radiation was used to investigate the element composition and valence state of the sample. The surface areas of the samples were determined by the Brunauer–Emmett–Teller (BET) method, and their pore size distributions were determined by the desorption branch of the isotherm using the Barrett–Joyner–Halenda (BJH) algorithm. The N_2_ sorption isotherms were collected using an ASAP 2020 (Micromeritics Inc., Norcross, GA, USA) system at a temperature of 77 K. The reduction behaviors and the interaction between the active phase and the support of each sample were examined by using the H_2_ temperature-programmed reduction (H_2_-TPR) technique. The H_2_-TPR measurements were performed with an Auto ChemTM II 2920 Instrument (Micromeritics Inc., Norcross, GA, USA) catalyst characterization system. H_2_-TPR was performed in a quartz cell, and approximately 50 mg of the sample was used for the measurement. The test temperature ranged from 50 to 850 °C with a heating rate of 10 °C/min.

### 2.3. Evaluation of the CO Oxidation Performance

The catalytic CO oxidation performance of the Pt/CeO_2_ catalyst was measured in a fixed-bed flow reactor with an internal diameter of 9 mm configured in a tube furnace. In a typical experiment, approximately 100 mg of the Pt/CeO_2_ catalyst was placed in the middle of the reactor supported by a quartz wool plug. The reaction gas mixture consisted of 1% CO, 10% O_2_ and 89% N_2_ (volume fraction). The total reaction gas flow was 100 mL min^−1^, and the corresponding space velocity was 60,000 h^−1^. The outlet gas was analyzed by an 7890B gas chromatograph equipped with a hydrogen flame detector (FID) (Anglit, Palo Alto, CA, USA). The CO conversion for CO oxidation can be calculated as follows:(1)$$\mathrm{CO}\;\mathrm{conversion}=\frac{C_{in}-C_{out}}{C_{in}}\times 100\%$$
where *C_in_* was the initial concentration of CO at room temperature and *C_out_* was the corresponding concentration of CO at the outlet as the temperature was increased.

## 3. Results 

Figure 1 shows the XRD patterns of dealloyed Al_91.9_Ce_8_Pt_0.1_, Al_91.7_Ce_8_Pt_0.3_ and Al_91.5_Ce_8_Pt_0.5_ ribbons calcined at different temperatures. As shown in Figure 1a, the diffraction peaks of the three samples can be indexed to the spinel cubic phase of CeO_2_ (PDF # 89-8436). However, no diffraction peaks corresponding to Pt were detected in the dealloyed and calcined Al_91.9_Ce_8_Pt_0.1_ sample; the intensity of the Pt peak in the diffraction patterns increased with increasing Pt content of the precursor alloys. The crystallinity of the dealloyed Al_91.7_Ce_8_Pt_0.3_ sample, as shown in Figure 1b, increased with increasing calcination temperature. However, the intensity of the Pt peak did not become stronger as the Pt content in the precursor alloys increases, which might be attributed to the high dispersity of the Pt.

The surface morphologies of the dealloyed and calcined Al_91.9_Ce_8_Pt_0.1_, Al_91.7_Ce_8_Pt_0.3_ and Al_91.5_Ce_8_Pt_0.5_ ribbons are shown in Figure 2. After dealloying and calcining, the CeO_2_ nanorods were formed and arranged in a framework structure with pores and holes distributed between the nanorods, as shown in Figure 2a,d,g. In this work, the diameter of the nanorods was approximately 20 nm in all the samples, indicating a small effect from the Pt content. At lower magnifications, the Pt/CeO_2_ appeared as a bulk material (insert of the figure). The TEM images of the Pt/CeO_2_ catalysts are shown in Figure 2b,e,h. When the Pt content is very low (Pt_0.1_/CeO_2_), it is difficult to observe the existence of Pt nanoparticles due to the low Pt content, as shown in Figure 2b. As observed from the SAED image in Figure 2c, only the diffraction ring of CeO_2_ can be observed, further indicating the low content and poor crystallinity of Pt on the surface of the CeO_2_ nanorod. When the Pt content in the precursor alloys increased to 0.3%, the size of the Pt nanoparticles was approximately 3–4 nm, as shown in Figure 2e. However, it is difficult to observe the diffraction ring of Pt from the selected area electron diffraction (SAED) image in Figure 2f, implying the existence of amorphous Pt. When the Pt content increased to 0.5%, aggregation of nanoparticles occurred, as observed from the yellow circle in Figure 2h. Furthermore, as clearly observed from Figure 2i, the reflection from (111) plane corresponded closely to Pt, which is consistent with the XRD results in Figure 1. 

To further prove the distribution of Pt in the Pt/CeO_2_ catalysts, STEM imaging and EDS mapping of the dealloyed Al_91.7_Ce_8_Pt_0.3_ ribbon calcined at 300 °C were conducted. The results indicate that the Pt nanoparticles were supported in situ and highly dispersed on the surface of the CeO_2_ nanorods, as shown in Figure 3. The nanoparticles had an average size of 3.5 nm, which is consistent with the TEM results above. In our previous works [31,32], Au/CeO_2_ catalysts with nanorod framework structures were prepared, and the formation of the catalysts was analyzed in detail. In this work, Pt/CeO_2_ catalysts with similar structures were obtained by dealloying and calcining melt-spun Al–Ce–Pt ribbons, and their formation mechanism was similar.

The specific surface area and pore size distribution of Pt/CeO_2_ nanorod framework were characterized by N_2_ adsorption–desorption tests. The N_2_ sorption isotherms and Barrett–Joyner–Halenda (BJH) pore size distributions are presented in Figure 4. The N_2_ sorption isotherms were type IV isotherms, with a hysteresis loop occurring at relative pressures ranging from 0.6–1.0 P/P_0_, as shown in Figure 4a, demonstrating the presence of a mesoporous structure [33]. The Pt_0.1_/CeO_2_, Pt_0.3_/CeO_2_ and Pt_0.5_/CeO_2_ nanorod frameworks had Brunauer–Emmett–Teller (BET) surface areas of 148.13, 161.95 and 148.44 m^2^ g^−1^, respectively, and the corresponding total pore volumes were 0.352, 0.3486 and 0.3264 cm^3^ g^−1^, respectively. The pore sizes for the Pt_0.1_/CeO_2_, Pt_0.3_/CeO_2_ and Pt_0.5_/CeO_2_ samples were centered around 8.88, 8.73 and 9.19 nm, respectively. Among them, the dealloyed and calcined Al_91.7_Ce_8_Pt_0.3_ sample exhibited the highest specific surface area and lowest pore diameter.

Figure 5 shows the catalytic performance of the Pt/CeO_2_ nanorod framework-structured catalysts with different Pt contents towards CO catalytic oxidation. For pure CeO_2_ prepared from dealloying Al_92_Ce_8_ alloys, the conversion temperatures for catalytic CO oxidation at 50% (T_50_) and 99% (T_99_) levels were 235 and 320 °C, respectively. The catalytic CO-oxidized performances were markedly improved upon the addition of Pt in the Al–Ce precursor alloys. For the Pt_0.1_/CeO_2_ catalyst, the temperatures for conversion of CO oxidation at 50% (T_50_) and 99% (T_99_) levels were 105 and 120 °C, respectively. As the Pt content was increased to 0.3% in the precursor alloy, the temperatures for conversion of CO oxidation at 50% (T_50_) and 99% (T_99_) levels were reduced to 91 °C and 113 °C, respectively. After the Pt content was increased to 0.5% in the precursor alloy, the total conversion temperature (T_99_) was 128 °C, and the catalytic activity followed a downward trend, as shown in Figure 5a. The temperature difference between T_50_ and T_99_ was only 22 °C for the Pt_0.3_/CeO_2_ catalyst, exhibiting a high thermal stability and an outstanding conversion rate at high temperatures. This means that the Pt/CeO_2_ catalyst exhibited a better square curve for the degree of CO conversion compared with that of the Au/CeO_2_ catalyst in our previous work [32]. Figure 5b shows the effect of the calcination temperature on the CO conversion of the catalysts prepared from the Al_91.7_Ce_8_Pt_0.3_ alloy. As observed, the total conversion temperatures (T_99_) for Pt_0.3_/CeO_2_ without calcination, and calcined at 200 °C, 300 °C, 400 °C, 500 °C were 170 °C, 113 °C, 113 °C, 117 °C, 126 °C, respectively. The conversion rates of the composites improved slightly with increasing calcination temperature from room temperature to 300 °C and then decreased. In comparison, the best catalytic performance was obtained by annealing at 200 °C and 300 °C, which resulted in complete conversion temperatures of approximately 113 °C.

The long-term stability of the Pt_0.3_/CeO_2_ catalyst was evaluated, and the results are shown in Figure 6. The Pt_0.3_/CeO_2_ catalyst exhibited nearly 98% CO conversion without noticeable deterioration in activity for a holding time of 75 h at 110 °C under a mixed composition of 1% CO, 10% O_2_ and 89% N_2_. To examine the long-term stability and the water resistance of the catalysts, CO conversion as a function of the holding time and the environmental effects was measured. The results reveal that the catalyst remained stable for 130 h when the reaction gas was mixed with 10% H_2_O. Typically, the addition of CO_2_ will cause adsorption competition with CO on the surface of Pt nanoparticles or the boundaries Pt and oxide, leading to a reduction in adsorption of activated CO per unit time, thus resulting in a decrease in catalytic CO oxidation activity [34,35,36]. However, the CO conservation could still remain stable when 5% CO_2_ and 10% H_2_O were introduced into the reaction gas after 65 h. This result implies that the nanorod framework structure in the present work is conducive to maintaining a high stability and excellent water resistance, even with the existence of adsorption competition.

Figure 7 shows an SEM image of the dealloyed Al_91.7_Ce_8_Pt_0.3_ calcined at 300 °C after continuous reaction for 130 h. The results show that the porous structure of the sample surface was well maintained and can clearly be observed after long-term testing. This result further demonstrates that the nanorod framework structure is conducive to maintaining a high stability.

X-ray photoelectron spectroscopy (XPS) was characterized to investigate the surface chemical states of the three catalysts. As shown in Figure 8a, the Ce 3d spectra of the Pt/CeO_2_ catalysts fitted into eight peaks, which are consistent with the peaks reported in the literature. The six peaks at 881.9 eV, 888.5 eV, 897.6 eV, 900.4 eV, 906.6 eV, and 916.1 eV can be assigned to Ce^4+^ [27]; while the other two peaks at 883.9 eV and 901.9 eV can be ascribed to Ce^3+^ [37,38]. Additionally, as calculated from the results of the Ce 3d spectra, the Ce^3+^ content of Pt_0.3_/CeO_2_ was 21.7%, which was a little higher than that of Pt_0.1_/CeO_2_ (21.2%) and Pt_0.5_/CeO_2_ (20.0%), indicating a higher concentration of oxygen vacancy on the surface of the Pt_0.3_/CeO_2_ sample [32,39]. Figure 8b shows that the ratios of Pt^0^/(Pt^0^ + Pt^2+^) for the Pt_0.1_/CeO_2_, Pt_0.3_/CeO_2_ and Pt_0.5_/CeO_2_ catalysts are 65.8%, 64.4% and 62.1%, respectively. For the Pt 4f spectrum of Pt_0.1_/CeO_2_, the peaks at 70.8 eV and 74.1 eV are attributed to the Pt^0^ species, and the peaks at 71.5 eV and 74.8 eV are attributed to the Pt^2+^ species. For the Pt 4f spectrum of Pt_0.3_/CeO_2_, the peaks at 70.8 eV and 74.1 eV are attributed to the Pt^0^ species, and the peaks at 72.0 eV and 76 eV are attributed to the Pt^2+^ species. Similarly, for the Pt 4f spectrum of Pt_0.5_/CeO_2_, the peaks at 70.7 eV and 74.0 eV are ascribed to the Pt^0^ species, and the peaks at 71.7 eV and 76.1 eV are attributed to the Pt^2+^ species [40].

Temperature-programmed reduction (TPR) by H_2_ was extensively used to characterize the oxygen reducibility of the dealloyed and calcined Al–Ce–Pt catalysts. Three distinct hydrogen consumption peaks can be observed in Figure 9. For Pt_0.1_/CeO_2_, Pt_0.3_/CeO_2_ and Pt_0.5_/CeO_2_, the first reduction peak is located at 101 °C, 80 °C, 116 °C, respectively. The second peak is located at 486 °C, 403 °C, 481 °C, respectively. The third peak is located at approximately 740 °C.

## 4. Discussions

The SEM and TEM images indicate the framework structures of the catalysts, with Pt nanoparticles of 3.5 nm in size being highly dispersed on the surface of the nanorods. The STEM analysis also proved that the Pt nanoparticles are formed in situ, supported and highly dispersed onto the surface of the CeO_2_ nanorods. This structure not only reflects the high catalytic activity of the Pt nanoparticles (Figure 5) but can also inhibit the agglomeration and growth of Pt particles during the catalytic process (Figure 7), thus contributing to the enhanced catalytic performance. The results of the BET and BJH analysis indicate that the dealloyed and calcined Al_91.7_Ce_8_Pt_0.3_ sample exhibited the highest specific surface area and lowest pore diameter among the three catalysts. Generally, a high specific surface area can provide an abundance of active catalytic sites for CO oxidation, while a good mesoporous structure provides more gas diffusion channels for the catalytic reaction [30]. In comparison, the dealloyed and calcined Al_91.7_Ce_8_Pt_0.3_ sample exhibited the highest specific surface area and lowest pore diameter among the three samples, thus exhibiting the best catalytic performance in the present work. The XPS results confirm that the Pt_0.3_/CeO_2_ catalyst has higher proportions of Ce^3+^ than those of the Pt_0.1_/CeO_2_ and Pt_0.5_/CeO_2_ catalysts, indicating its higher concentration of oxygen vacancies and active sites. The oxygen vacancies can induce the adsorption of oxygen species with low-temperature adsorption–desorption properties and can promote the transfer of electrons between metal ions as well as the activity of the lattice oxygen; the active sites can provide more reaction paths for catalytic CO oxidation, thereby enhancing the catalytic ability. For the H_2_-TPR result in Figure 9, the appearance of the first peak for Pt_0.1_/CeO_2_, Pt_0.3_/CeO_2_ and Pt_0.5_/CeO_2_ catalysts is due to Pt-oxide reduction, which demonstrates that the Pt_0.3_/CeO_2_ sample shows better adsorption ability for reduction gas. The occurrence of the second peak is ascribed to the reduction of surface-active oxygen. The reduction peak for the Pt_0.3_/CeO_2_ sample stays left, and its integrated area of the peak is much larger than that of the other two samples, implying that the content of surface-active oxygen on CeO_2_ for Pt_0.3_/CeO_2_ is significantly higher than that of the other two samples. The amount of surface-active oxygen is also an important element in the evaluation of the catalytic ability of a catalyst, thus further demonstrating that Pt_0.3_/CeO_2_ possesses the best catalytic activity. The third peak at approximately 740 °C is due to the existence of CeO_2_ in the catalyst [41,42,43]. Durability testing of the Pt_0.3_/CeO_2_ catalysts indicated that the catalyst had a good ability, which may be explained by the high catalytic activity of the in situ supported Pt nanoparticles and the thermal and catalytic stability of the nanorod framework structure. The CeO_2_ catalysts can effectively prevent the deposition of carbon and lead to the formation of fewer by-products during the catalytic process.

A comparison of the nanorod framework-structured Pt/CeO_2_ catalysts with different reported Pt/CeO_2_ catalysts is provided in Table 1. The results show that the nanorod framework structure of the Pt/CeO_2_ catalysts in the present paper exhibited a high catalytic activity and an excellent stability compared to those of previous reports.

The corresponding schematic illustration of the CO oxidation mechanism is shown in Figure 10. The oxygen vacancies, which play a key role in CO catalytic oxidation, are mainly generated at the Pt–CeO_2_ interface or boundary. Initially, CO is mainly adsorbed onto the Pt surface and reacts with the oxygen species activated by the oxygen vacancies at the Pt–CeO_2_ boundary, leading to the formation of CO_2_ and more surface oxygen vacancies. The strongly adsorbed CO reacts with the active oxygen species adsorbed at the surface vacancies [46]. The existence of the Pt particles can accelerate the migration of active oxygen adsorbed onto the CeO_2_ nanorod. The migration of the generated oxygen vacancies provides access to neighboring surface lattice oxygen for the following cycles [32], contributing to the high and efficient CO conversion. This result conforms to the CeO_2_-assisted Mars–van Krevelen mechanism [46].

## 5. Conclusions

In summary, the nanorod framework-structured Pt/CeO_2_ catalysts were fabricated by a simple dealloying method. The SEM and TEM analysis indicated the framework structure of the catalysts, with Pt nanoparticles of 3.5 nm in size highly dispersed over the surface. The catalytic tests showed that the T_99_ temperatures of CeO_2_, Pt_0.1_/CeO_2_, Pt_0.3_/CeO_2_, Pt_0.5_/CeO_2_ were 320 °C, 120 °C, 113 °C, 128 °C, respectively. Obviously, the Pt/CeO_2_ catalysts exhibited enhanced catalytic performance towards CO oxidation compared with CeO_2_, which is ascribed to their high specific surface area and the uniform distribution of Pt particles. XPS and H_2_-TPR results showed that the Pt_0.3_/CeO_2_ sample exhibited a higher Ce^3+^ cation concentration, richer oxygen vacancy and better adsorption ability for reduction gas, thus showing the best catalytic activity. The Pt_0.3_/CeO_2_ sample showed the highest catalytic activity when calcined at 300 °C. Moreover, the catalyst could remain stable for 130 h when water was added to the reaction gas, demonstrating the excellent long-term stability and strong water resistance of the catalyst.

## Figures and Tables

**Figure 1 nanomaterials-09-00683-f001:**
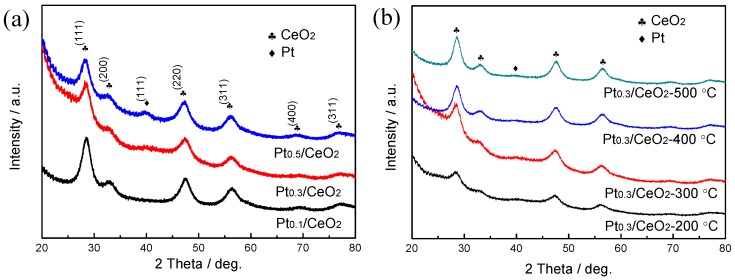
XRD patterns of (**a**) dealloyed Al_91.9_Ce_8_Pt_0.1_, Al_91.7_Ce8Pt_0.3_ and Al_91.5_Ce_8_Pt_0.5_ ribbons calcined at 300 °C and (**b**) dealloyed Al_91.7_Ce_8_Pt_0.3_ ribbons calcined at 200, 300, 400 and 500 °C.

**Figure 2 nanomaterials-09-00683-f002:**
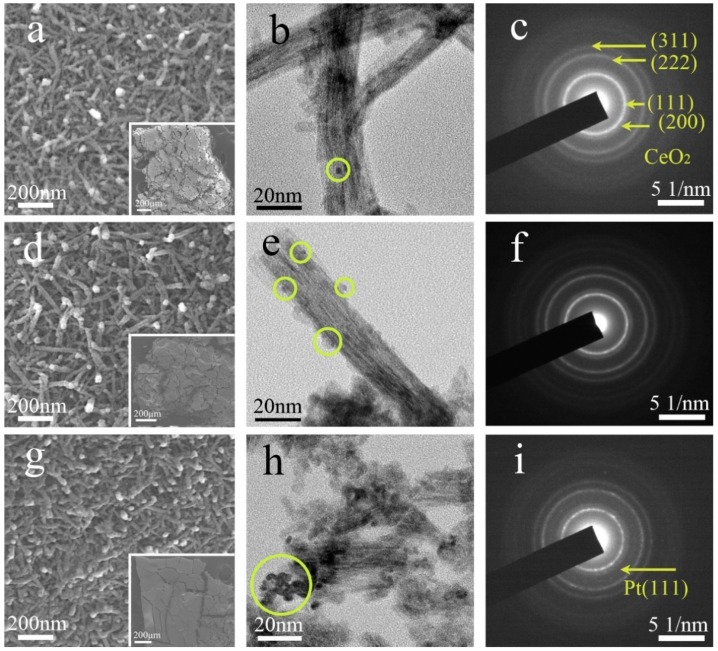
SEM, TEM and selected area electron diffraction (SAED) images of (**a**–**c**) dealloyed Al_91.9_Ce_8_Pt_0.1_, (**d**–**f**) Al_91.7_Ce_8_Pt_0.3_ and (**g**–**i**) Al_91.5_Ce_8_Pt_0.5_ ribbons calcined at 300 °C. The insert images of (**a**,**d**,**g**) are the low magnification SEM images of the samples.

**Figure 3 nanomaterials-09-00683-f003:**
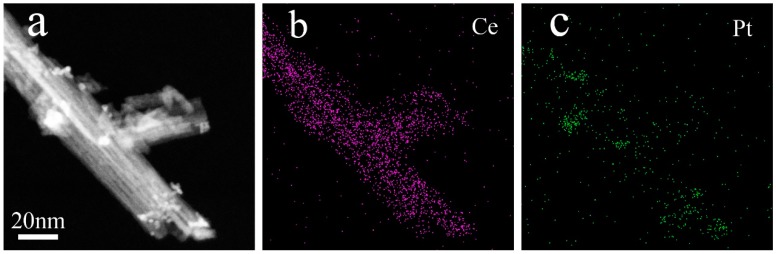
(**a**) Scanning transmission electron microscopy (STEM) image and element mapping of (**b**) Ce and (**c**) Pt in the Pt/CeO_2_ catalyst prepared from the dealloyed Al_91.7_Ce_8_Pt_0.3_ ribbon calcined at 300 °C.

**Figure 4 nanomaterials-09-00683-f004:**
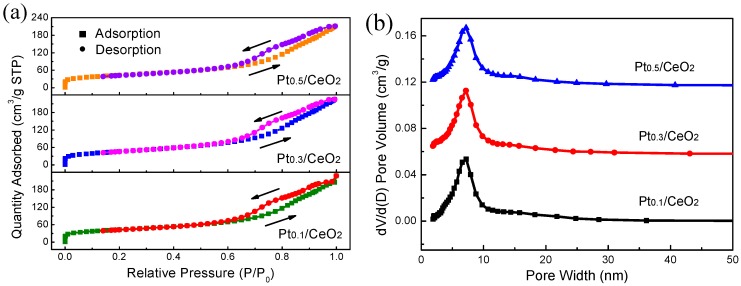
(**a**) Nitrogen adsorption–desorption isotherms and (**b**) the Barrett–Joyner–Halenda (BJH) pore size distributions of dealloyed Al_91.9_Ce_8_Pt_0.1_, Al_91.7_Ce_8_Pt_0.3_ and Al_91.5_Ce_8_Pt_0.5_ nanorod frameworks calcined at 300 °C.

**Figure 5 nanomaterials-09-00683-f005:**
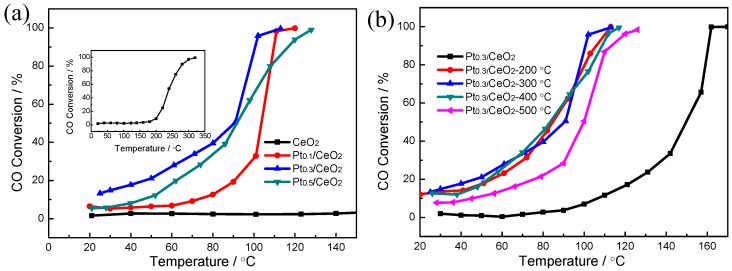
(**a**) CO conversion as a function of reaction temperature over the CeO_2_, Pt_0.1_/CeO_2_, Pt_0.3_/CeO_2_ and Pt_0.5_/CeO_2_ catalysts; (**b**) the Pt_0.3_/CeO_2_ catalyst prepared from dealloyed Al_91.7_Ce_8_Pt_0.3_ calcined at different temperatures. The inset image of (**a**) is the complete conversion curve for CeO_2_.

**Figure 6 nanomaterials-09-00683-f006:**
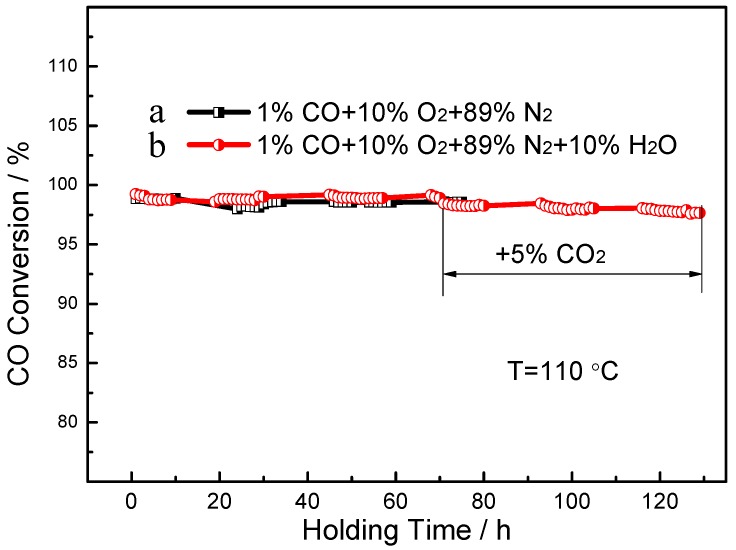
Long-term stability of the Pt_0.3_/CeO_2_ catalyst tested at 110 °C. (**a**) Successive reaction under a mixed composition of 1% CO, 10% O_2_ and 89% N_2_ for 75 h; (**b**) Successive reaction under mixed composition of 1% CO, 10% O_2_, 79% N_2_ and 10% H_2_O for 130 h, with addition of 5% CO_2_ after reaction for 70 h.

**Figure 7 nanomaterials-09-00683-f007:**
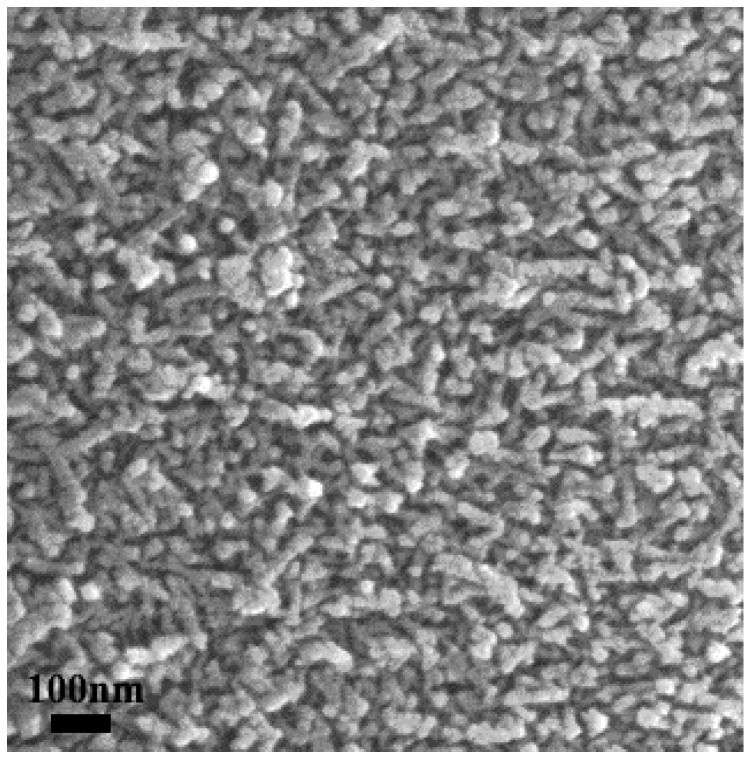
SEM image of the Pt_0.3_/CeO_2_ catalyst after reaction for 130 h.

**Figure 8 nanomaterials-09-00683-f008:**
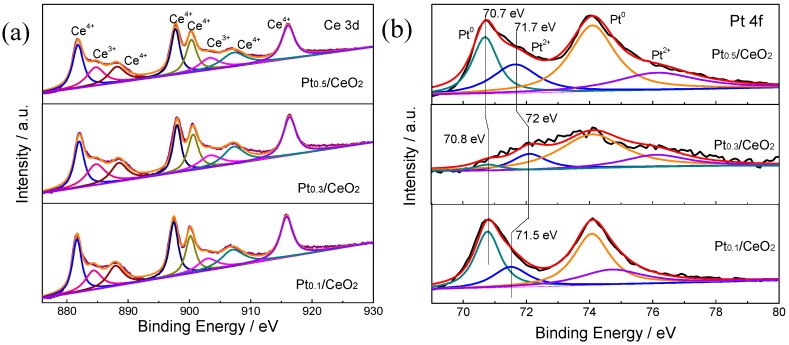
X-ray photoelectron spectroscopy (XPS) spectra of (**a**) the Ce 3d and (**b**) the Pt 4f peaks for the Pt_0.1_/CeO_2_, Pt_0.3_/CeO_2_ and Pt_0.5_/CeO_2_ catalysts.

**Figure 9 nanomaterials-09-00683-f009:**
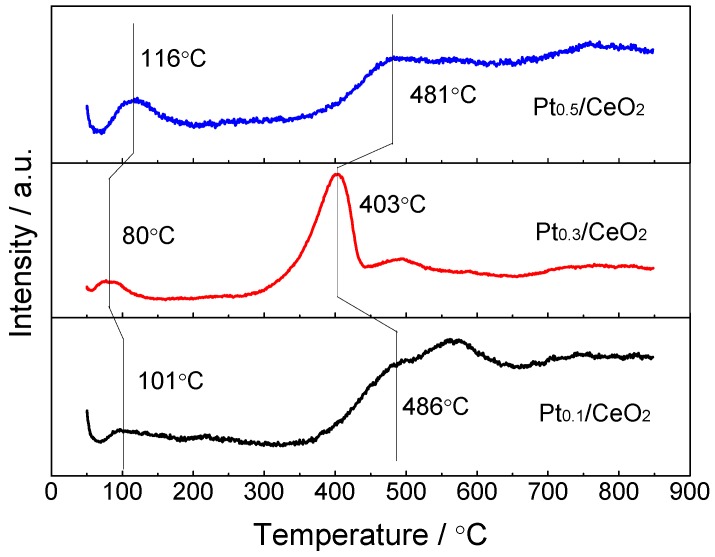
H_2_ temperature-programmed reduction (H_2_-TPR) profile of Pt_0.1_/CeO_2_, Pt_0.3_/CeO_2_ and Pt_0.5_/CeO_2_ catalysts.

**Figure 10 nanomaterials-09-00683-f010:**
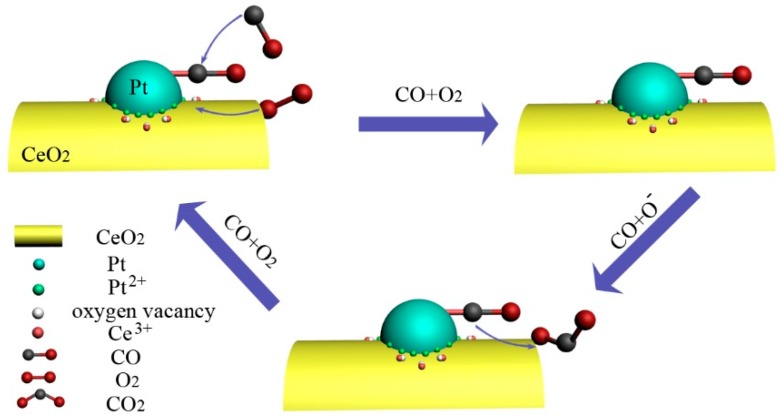
CO oxidation reaction mechanism on the Pt/CeO_2_ nanorod catalysts.

**Table 1 nanomaterials-09-00683-t001:** Comparison of the catalytic performance of the nanorod framework-structured Pt/CeO_2_ catalysts with those of previous reports.

Catalyst	Reaction Temperature (T_99_) (°C)	Durability Test (h)	Space Velocity (h^−1^)	Reference
Porous/hollow structured Pt/CeO_2_@SiO_2_	162	Not stated	60,000	[44]
Nanorods CeO_2_-IMP-Pt	90	10	30,000	[45]
Pt/CeO_2_ hollow sphere	155	12	80,000	[25]
Pt/CeO_2_ mesoporous sphere	95	12	80,000	[14]
Nanorod framework-structured Au/CeO_2_	90	Not stated	60,000	[32]
Nanorod framework-structured Pt_0.3_/CeO_2_	113	130	60,000	This work

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
