# Peer review of "The Preparation and Catalytic Properties of Nanoporous Pt/CeO2 Composites with Nanorod Framework Structures"

_nanomaterials, 2019, doi:10.3390/nano9050683_

Reviewer 1 Report

This paper by Sun et al describes the preparation, characterization, and catalytic performance of novel Pt/CeO2 composites with porous structure. The composites are well characterized by several technique including XRD, TEM, and XPS measurements, and catalytic performance is also nicely investigated. They also discussed the effect of Pt contents on the catalytic activity of the composites. Overall, the contents described in the manuscript is well supported by the experimental evidence and the significance of the work is appropriately described. Therefore, this reviewer recommends the publication of this manuscript as the current form. 

Author Response

Dear Editor:

Thank you for your letter and the reviewers’ comments concerning our manuscript entitled “The preparation and catalytic properties of nanoporous Pt/CeO2 composites with nanorod framework structures” (ID: nanomaterials-482121). Those comments are valuable and very helpful for revising and improving the manuscript, as well as the important guiding significance to our researches. According to the reviewers’ detailed suggestions, we have made a careful revision on the original manuscript. The revised are marked in yellow in the manuscript. The main corrections and the responds to the reviewers’ comments are listed in the attachment.

We tried our best to improve the manuscript and made some changes in the manuscript. These changes will not influence the content and frame work of the manuscript. And here we did not list the changes but marked in blue in the revised. We appreciate Editors/Reviewers’ warm work earnestly, and hope that the correction will meet with approval. Once again, thank you very much for your comments and suggestions. 

Best wishes,

Sincerely yours!

Reviewer 2 Report

The manuscript is clearly written. 

The characterization of the new material is carried out carefully..

The catalytic performance is satisfactory.

Overall the quality of the manuscript and importance of the results obtained merits its publication.

Author Response

(The authors gave the same response as above.)

Reviewer 3 Report

Referee of the paper: The preparation and catalytic properties of nanoporous Pt/CeO2 composites with nanorod framework structures

The manuscript presents a study of Pt/CeO2 nanocomposites studied for CO oxidation. Fresh samples were characterized by XRD, BET, TPR (only one), XPS, SEM, TEM, STEM (only one). Only one of used samples was characterized by XPS (recorded after air exposition, comporting surface re-oxidation).

The paper present a lot of criticisms: it is confused in the exposition, lacks in experimental information, not deepen in characterization, elaboration and discussion of XPS appear critical and not correct. Some catalytic curves do not have good quality. Some claims are not justified by the results. Catalytic curves do not have quality to be published.

The main criticisms:

1) abstract mentions only some of the techniques used in structural characterization.

2) Experimental are not complete: TPR and XPS are not mentioned at all in experimental.

3) TPR and XPS of the used sample are in the discussion paragraph instead of in the results paragraph.

4) Fig 5c e 5d show catalytic data completely not mentioned, nor described and nor discussed in the text and totally not present in the caption

5) curves of catalytic activity in fig 5 does not show to zero conversion at low T. In particular Ce8Pt0.5 in fig 5a and Ce8Pt0.3-400 °C Ce8Pt0.3-500 °C in fig 5b. These last two curves show a linear activity of 30% instead of 0% at low temperature! How the authors can explain that? These data do not have good quality, probably due to errors in the acquisition data or GC analysis. They have not the quality for to be published and have to be performed again.

Fig 5a and 5b show a wide temperature range not interesting. They shoud be limited to 0-150 °C and give more evidence to the activity range of Pt catalysts.

6) line 188. the addition of CO2: why authors added CO2?. The aim of the addition is not clear and should be explained.

7) XPS analysis and discussion present serious problems:

A) Pt 0.1 deconvolution spectra have to show the same color for the same species of other samples.

B) Line 2015: Additionally, as calculated from the fits of the Ce 3d spectra, the Ce3+ content of Pt0.3/CeO2 is 23.7%, which is much higher than that of Pt0.1/CeO2 (21.2%) and Pt0.5/CeO2 (20.7%), indicating a higher concentration of oxygen vacancy on the surface of the Pt0.3/CeO2 sample.

A difference between 23,7 and 21.2-20,7 it is not a MUCH HIGHER . It is hazardous from authors attribute that importance to a deconvolution of a spectrum (Pt 0.3) of so bad quality, that is not so certain in deconvolution and area of V’ component.

C) Line 208: Fig. 8(b) shows that the ratios of Pt0/(Pt0 + Pt2+) for the Pt0.1/CeO2, Pt0.3/CeO2 and Pt0.5/CeO2 catalysts are 65.8%, 66.4% and 62.1%, respectively, based on the calculated compositions of the different Pt species; a higher amount of Pt0 leads to a stronger CO adsorption capacity, indicating the high activity of Pt0.3/CeO2.

Line211. spectrum of Pt0.3/CeO2, the peaks at 71.8 eV and 75.2 eV are attributed to the Pt0 species, and the peaks at 72.5 eV and 75.7 eV are attributed to the Pt2+ species.

The different position of peaks for sample 0,1 and 0.5 are not detailed in the text but only showed in the figure at very different position. The deconvolution of spectra should show the same components at nearly the same position. How authors could justify the position so different in the sample 0.3Pt that is similar in composition and in preparation? That attribution appear not convincing at all.

The deconvolution of Pt 0.3 seem clearly do not have the components of Pt0 but only a great amount of Pt+2. That presence of Pt+2 instead of Pt0 should be the most important feature to discuss in order to explain the higher activity. Authors have to take in serious account this comment.

8) In addition the reader has not any idea of how the XPS experiment were conducted.

9) Fig 9 is a miscellaneous of data and techniques: TPR is relative of calcined sample, XPS is relative of used sample. TPR have to be moved in the previous paragraph. TPR is relative only to one sample and it does not give any useful information, it should be completed with the other sample or removed

10) line 260. Furthermore, it can be observed that the ratio of Pt2+ increases, implying that Pt2+ is produced during the reaction.

Authors should better show the comparison: between which components??? Which spectra?? . The profile in fig )c is centered at 74 eV. Whereas in fig 8b is centered at 75 eV. Is the same sample???? And why the peak is now at 74?

11) Fig 9b and 9c:  samples "used" but the treatment reported is CO + N2 without O2…    is that the treatment? In that case, the sample is not really used in CO oxidation.

12) Line 280  Obviously, the Pt0.3/CeO2 catalyst exhibits enhanced catalytic performance towards CO oxidation compared with the other dealloyed samples, which is ascribed to its high specific surface area and uniform distribution of Pt particles.

High specific surface area is feature of all the three samples. Huniform distribution of Pt by STEM was performed only for that sample. Authors do not do any hypotesys about the effect of the studied parameters such as calcination temperature or Pt content on the catalytic activity. In addition authors ignore completely the discussion of XPS results.

Minor revisions:

1) Fig 3c. The stem of in color blue is not visible. Change the color.

2) line 137: inserts are not  magnified SEM

3) capion fig. 6: add experimental conditions

Author Response

Dear Editor:

Thank you for your comments concerning our manuscript entitled “The preparation and catalytic properties of nanoporous Pt/CeO2 composites with nanorod framework structures (ID: nanomaterials-482121)” . Those comments are valuable and very helpful for revising and improving the manuscript, as well as the important guiding significance to our researches. According to the reviewers’ detailed suggestions, we have made a careful revision on the original manuscript. The revised are marked in yellow in the manuscript. The main corrections and the responds to the reviewers’ comments are listed in the attachment.

We tried our best to improve the manuscript and made some changes in the manuscript. These changes will not influence the content and frame work of the manuscript. And here we did not list the changes but marked in blue in the revised. We appreciate Editors/Reviewers’ warm work earnestly, and hope that the correction will meet with approval. Once again, thank you very much for your comments and suggestions. 

Best wishes,

Sincerely yours!

Round  2

Reviewer 3 Report

The authors accepted all the suggestions. The paper is much improved and can now be published